



# Spectral dependence of birch and pine pollen optical properties using a synergy of lidar instruments

Maria Filioglou[1], Ari Leskinen[1,2], Ville Vakkari[3,4], Ewan O'Connor[3], Minttu Tuononen[5], Pekko Tuominen[5], Samuli Laukkanen[5], Linnea Toiviainen[6], Annika Saarto[6], Xiaoxia Shang[1], Petri Tiitta[1], and Mika Komppula[1]

[1]Finnish Meteorological Institute, Kuopio, Finland
[2]Department of Technical Physics, University of Eastern Finland, Kuopio, Finland
[3]Finnish Meteorological Institute, Helsinki, Finland
[4]Atmospheric Chemistry Research Group, Chemical Resource Beneficiation, North-West University, Potchefstroom, South Africa
[5]Vaisala Oyj, Vantaa, Finland
[6]The Biodiversity Unit of the University of Turku, Turku, Finland

**Correspondence:** Maria Filioglou (maria.filioglou@fmi.fi)

**Abstract.** Active remote sensors equipped with the capability to detect polarization, a shape relevant parameter, are essential to aerosol particle identification in the vertical domain. Most commonly, the linear particle depolarization ratio has been available at the shorter wavelengths of 355 nm and/or 532 nm. Recently, linear particle depolarization ratios at longer wavelengths (910 nm, 1064 nm and, 1565 nm) have emerged to the lidar aerosol research. In this study, a synergy of three lidars, namely a PollyXT lidar, a Vaisala CL61 ceilometer and a Halo Photonics StreamLine Pro Doppler lidar, and in situ aerosol and pollen observations have been utilized to investigate the spectral dependence of birch and pine pollen particles. We found that regardless of the pollen type, the linear particle depolarization ratio was subject to the amount of pollen and its relative contribution to the aerosol mixture in the air. More specifically, during the birch pollination characteristic linear particle depolarization ratios of $5 \pm 2\%$ (355 nm), $28 \pm 6\%$ (532 nm), $23 \pm 6\%$ (910 nm) and, $33 \pm 4\%$ (1565 nm) were retrieved at the pollen layer. Regarding the pine dominant period, the characteristic linear particle depolarization ratio of $6 \pm 2\%$, $43 \pm 11\%$, $22 \pm 6\%$ and, $26 \pm 3\%$, was determined at 355 nm, 532 nm, 910 nm and, 1565 nm wavelengths, respectively. For birch, the linear particle depolarization ratio at 1565 nm was the highest followed by 532 nm and 910 nnm wavelengths, respectively. A sharp decrease at 355 nm was evident for birch pollen. For pine pollen, a maximum at 532 nm wavelength was observed. There was no significant change in the linear particle depolarization ratio at 910 nm for the pollen types considered in this study. Given the low concentration of pollen in the air, the inclusion of the longer wavelengths (910 nm and 1565 nm) for the detection of birch and pine can be beneficial due to their sensitivity to trace large aerosol particles.

## 1 Introduction

Pollen, a major mass contributor of atmospheric primary bioaerosols during pollination at the higher latitudes of the Northern Hemisphere (Manninen et al., 2014; Williams and Després, 2017), is closely associated to allergic diseases (Kitinoja et al.,





2020). The ongoing allergic disease increment in the population (Pawankar et al., 2008) is projected to further double by 2040-2060 in Europe alone (Lake et al., 2018), pacing pollen as one of the most important consequences of climate change for human health (Beggs, 2015). Alteration of pollen season timing and load is not only triggering allergy-related symptoms but it can potentially alter the climate. Although the direct radiative effect of pollen is likely globally small due to the low particle number concentration (Löndahl, 2014), its indirect effect is potentially relevant. Several studies have demonstrated

that pollen is moderately hygroscopic, acting as cloud condensation nuclei (CCN) under very low supersaturation (Pope, 2010; Griffiths et al., 2012; Prisle et al., 2019; Mikhailov et al., 2019). Pope (2010) suggested that intact pollen grains act as giant CCNs in cloud processing. In a laboratory study, Steiner et al. (2015) hypothesized that sub-micron sub-pollen particles (SPPs) are more likely to contribute to cloud processing compared to intact pollen grains. They concluded that SSPs can be a global CCN contributor depending on their concentration. Using an atmospheric pollen grain rupture model, Wozniak et al. (2018)

simulated a suppress of seasonal precipitation of about 30% in the presence of SPPs in clean continental aerosol conditions. In addition to its CCN activity, intact pollen grains and fragments of it are an effective ice nuclei (IN), having the ability to form ice crystals at warm subfreezing temperatures. (Diehl et al., 2001, 2002; Pummer et al., 2012; Hader et al., 2014; Dreischmeier et al., 2017; Gute et al., 2020; Burkart et al., 2021). Consequently, monitoring pollen concentration and its distribution in the atmosphere is essential.

Continuous monitoring of pollen near ground level is usually performed using Hirst-type volumetric air samplers (Buters et al., 2018). Although, these instruments serve as the reference system in pollen monitoring, they cannot fulfil the high demand for real-time or near real-time information, and they are very labour intensive. Promising novel instrumentation emerges in pollen detection and enables the automated and continuous monitoring of pollen. These techniques utilize either image recognition (Sauvageat et al., 2020; Oteros et al., 2020) or the fluorescence spectra (Crouzy et al., 2016; Saito et al., 2018;

Richardson et al., 2019). However, systematic information on the vertical distribution and optical properties of pollen is not possible for most of the aforementioned approaches. Inclusion of the vertical information is not only essential to aerosol identification algorithms but also to model verification and assimilation studies.

Lidars have been increasingly utilized to study pollen optical properties since their non-spherical shape induce strong laser depolarization. The first evidence of the lidar capability to sense pollen dates back to 2008 where Sassen (2008) reported a

linear particle depolarisation ratio (PDR) of 30% at 694 nm wavelength in plumes of paper birch (*Betula papyrifera*) in Alaska, U.S.A. Noh et al. (2013a) and Noh et al. (2013b) studied the diurnal variability and the vertical distribution of pollen coupling the higher PDR values to higher number pollen concentrations during noon times. Using a Micro Pulse lidar at 532 nm, Sicard et al. (2016) observed diurnal mean (hourly maxima) PDR of 14 (33)% in a mixture of plane (*Platanus*), pine (*Pinus*) and cypress (*Cupressaceae*) pollen in Barcelona, Spain. In recent studies, multi-wavelength high-power lidars have shed further

light into the pollen optical properties. Using the Raman lidar technique, Bohlmann et al. (2019) retrieved mean lidar ratio (LR) of $45 \pm 7$ sr (355 nm) and $55 \pm 16$ sr (532 nm) in aerosol layers with birch presence in Finland. It became evident that in order to characterize the pure pollen optical properties, the contribution of other aerosol types has to be constrained as well. To this direction, Shang et al. (2020) and Shang et al. (2022) proposed a methodology to estimate the pollen PDR using the backscatter-related Ångström exponent (BÅE) assuming that pollen particles induce a BÅE of $0 \pm 0.5$ due to their



large size. They estimated the PDR of pure pollen particles at 532 nm to be $24 \pm 4\%$ and $36 \pm 1\%$ for Silver birch (*Betula pendula*) and Scots pine (*Pinus sylvestris*), respectively. No significant wavelength dependence in the LR at both 355 and 532 nm wavelengths was observed, with values ranging between 55 and 70 sr. The spectral dependence of pollen PDR was reported in the recent work of Bohlmann et al. (2021). Using a synergy of a multi-wavelength lidar and a Doppler lidar, a strong decrease in the PDR at 355 nm was observed compared to the longer wavelengths of 532 and 1565 nm, marking the

importance of two (or more) polarisation wavelengths for pollen identification. Veselovskii et al. (2021) demonstrated that a combination of PDR and fluorescence spectra can be utilized in aerosol classification schemes to distinguish bioaresols. As far as laboratory-based pollen optical properties concern, Cao et al. (2010) measured the PDR of various pollen types at four wavelengths (355, 532, 1064, and 1570 nm). They showed that each pollen type has a specific signature and this can be used to distinguish bioaerosols in the atmosphere. The latest studies of Cholleton et al. (2022a) and Cholleton et al. (2022b) concluded

that a difference in the scattering angle of just 2.5° (from 177.5° to 180°) induces a substantial relative error in the pollen PDR ranging from 20 to 40 %.

To this end, the majority of atmospheric lidar studies have reported the observed PDR for a narrow range of pollen concentrations and/or mixtures of these, focusing mainly on case studies. Since the value of the PDR in the presence of non-spherical or inhomogeneous particles depends on the amount, the complexity of particle shape (including its orientation) and the particle

size relative to the wavelength, its value can greatly fluctuate depending on the aerosol population. Therefore, both the amount and the relative share of pollen compared to the rest of aerosols in the volume has to be considered. In this study, two years of pollen observations, one with exceptionally high pollen concentration and one with moderate one, were utilized to constrain the characteristic optical properties of birch (*Betula*) and Scots pine (*Pinus sylvestris*). We present the wavelength dependence of the aforementioned pollen types considering co-located observations from three lidar instruments, namely, a PollyXT lidar,

a CL61 ceilometer and a Halo Photonics Streamline Pro Doppler lidar, all equipped with polarization capability. Combined with in situ pollen and aerosol observations, the characteristic pollen PDR and equivalent BÅE for each pollen type were determined.

## 2 Instrumentation and methods

Two field campaigns conducted during 2021 and 2022 at the forest site in Vehmasmäki (Kuopio), Finland (62°44′ N, 27°33′ E;

190 m above sea level). The rural station is surrounded by broad-leaved and coniferous trees. Birch (*Betula*), Scots pine (*Pinus sylvestris*), alder (*Alnus*), nettle (*Urtica*) and Norway spruce (*Picea abies*) are the dominant pollen types, nonetheless their concentration can greatly vary from year to another. On site, a multi-wavelength PollyXT lidar, a Vaisala CL61 ceilometer, a Halo Photonics Stremline Pro Doppler lidar and various in situ instruments for the aerosol characterization as well as meteorological quantities from a 318 m tall mast, were available. Furthermore, a Hirst-type volumetric air sampler was deployed to

measure the concentration of pollen during the pollination period which usually lasts from March to August each year.





## 2.1 The PollyXT lidar

The multi-wavelength PollyXT lidar (Engelmann et al., 2016) enables the determination of the particle backscatter coefficient at three wavelengths (355, 532 and 1064 nm) and the volume depolarization ratio (VDR) and PDR at two wavelengths (355 and 532 nm). Additionally, extinction coefficients at 355 and 532 nm are available during nighttime using the Raman technique. Using the 407 nm Raman-shifted wavelength, water vapor mixing ratio profiles can be also retrieved during dark hours (Filioglou et al., 2017). Apart from the far-field telescope, PollyXT features a near-field unit retaining full overlap at about 120 m. Data are recorded with a vertical resolution of 7.5 m and a temporal resolution of 30 s. Further details on the instrument, its operating principle and uncertainties can be found in Baars et al. (2012) and Engelmann et al. (2016).

## 2.2 The Vaisala CL61 ceilometer

The Vaisala CL61 ceilometer uses a pulsed laser diode at 910.55 nm to emit light towards the atmosphere. The single-channel lidar uses an alternating polarizing sheet filter which enables the retrieval of the attenuated backscatter coefficient and VDR/PDR. Full overlap is obtained at about 300 m. Profiles are available at a temporal resolution of 5 s (for the attenuated backscatter coefficient) and 10 s (for the VDR) and a vertical resolution of 4.8 m.

## 2.3 The Halo Photonics StreamLine Pro Doppler lidar

The Halo Photonics StreamLine Pro Doppler lidar (Pearson et al., 2009) operates at 1565 nm wavelength. The lidar can retrieve information down to 90 m above ground. It is equipped with a cross-polar receiver channel, enabling the determination of PDR at the same wavelength (Vakkari et al., 2021). The lidar has a vertical resolution of 30 m. Further operating specifications can be found at Vakkari et al. (2021).

## 2.4 Hirst-type volumetric air sampler

Sampling of airborne pollen was performed at 4 m above ground level using a Hirst-type volumetric air sampler (Hirst, 1952). The instrument samples air with a flow rate of $10 \, \mathrm{l \, min^{-1}}$. Aerosol particles impact onto a sticky tape mounted on a rotating drum having a 7-day cycle which later on is divided into 24 h segments and analyzed offline using light microscopy on 48 randomized fields with 400x magnification. An accuracy of 2 h is obtained with this sampling and analyzing methodology. The main sources of error for this instrument are variations in flow rate, the counting method and the different mounting media. Although these sources have been addressed and constrained, a recent study has shown that discrepancies between samplers are about 16 % when pollen concentration is above 10 pollen grains $\mathrm{m^{-3}}$ and further increases for lower pollen concentration (Adamov et al., 2021).

## 2.5 Aerosol in situ observations

Samples for measuring the aerosol particle size distribution were taken 5 m above ground level and delivered via a stainless steel line (3 m long, 8 mm inner diameter) to a Nanoscan scanning mobility particle sizer Model 3910 (NS) and an optical





particle sizer Model 3330 (OPS), both manufactured by TSI Incorporated. The inlet flow and measurement particle size range were 0.8 l min$^{-1}$ and 10-420 nm (mobility diameter) with 13 size bins for the NS, and 1.0 l min$^{-1}$ and 0.3-10 $\mu$m (optical diameter) with 16 size bins for the OPS. The time resolution for both instruments was 1 min.

Moreover, black carbon (BC) concentration and aerosol light scattering coefficient were measured with a Magee Scientific

Aethalometer Model AE-31 (AE) and a TSI nephelometer Model 3563 (NM), respectively. The AE outputs the BC concentration at seven wavelengths between 370-950 nm, which can be then converted to aerosol absorption coefficients. The NM outputs the aerosol scattering coefficient at the wavelengths of 450 nm, 550 nm, and 700 nm. The time resolution for both instruments was 5 min. Regarding the NM data analyses, a correction for the truncation error as in Anderson and Ogren (1998) using the no-cut regime was applied. The AE data were corrected for filter loading and multiple scattering following Leskinen

et al. (2020) methodology, using a long-term average correction factor of 4.75 at 880 nm for the multiple scattering at the Vehmasmäki station.

### 2.6    Methodology

Considering that the Hirst-type volumetric air sampler presents the coarsest temporal resolution, observations from all other instruments were temporally averaged to match the 2h resolution from the pollen sampler. In this study, we argue that the PDR

can fluctuate depending on the amount of pollen and other aerosols in the aerosol mixture. In order to calculate the share of pollen to other aerosols, we have converted the pollen and aerosol number concentration derived from the pollen sampler and in situ instruments to particulate matter (PM). The mass is a more comparable quantity for the lidar technique than the number concentration. For the conversion of the pollen particles from the pollen sampler, particle diameter of 25 and 75 $\mu$m and a particle density of 0.8 and 0.4 g cm$^{-3}$ for birch and pine pollen, respectively, was considered (Gregory, 1961).

To calculate the share of other aerosols in the aerosol mixture, the mass concentration of PM$_{10}$ was estimated using the in situ aerosol observations. In order to so do, the NS and OPS aerosol size distributions were combined. The NS aerosol size distribution and the first bin of the OPS aerosol size distribution were neglected due to observed inaccuracies in the data. The optical diameters applied in the OPS, calibrated with polystyrene latex particles with a refractive index of 1.59–0.00$i$, were converted to geometric mean volume equivalent diameters following the procedure in Alas et al. (2019) and by using a long-

term average refractive index of 1.46–0.009$i$ at the Vehmasmäki station. This long-term average was obtained from matching Mie calculated scattering and absorption coefficients to those measured with the NM and AE instruments. The measured number aerosol size distributions were then converted to volume aerosol size distributions assuming all particles as spherical. Then, the individual volume aerosol size distributions from NS and OPS instruments were combined to one. The values for the missing size bins in the merging region between NS and OPS instruments were calculated by fitting a log-normal size

distribution to each measured size distribution. The final volume size distribution, which had 31 size bins, was then multiplied by a size-independent particle density of 1.6 g cm$^{-3}$ and integrated between 0.01 and 10 $\mu$m in order to estimate the mass concentration of PM$_{10}$.

Regarding the retrieval of the optical profiles from PollyXT lidar, the backward Klett inversion (Klett, 1981) was applied to the 2h temporally averaged signals. Unless otherwise indicated by the Raman retrieval, LRs between 55 and 65 sr were utilized.



Considering the near field capability and the vertical smoothing, profiles down to 400 m above ground were considered. Below this height, all retrieved optical properties were assumed constant and equal to the value at 400 m. This was necessary not to bias the mean optical properties of the first layer which is presumed to contain the pollen particles. Note that due to an overlap issue at 355 nm far-range channel, reliable profiles at this wavelength for the PDR were limited to 800 m above ground. To ensure high-quality PDR retrievals, the instrument performs the $\Delta 90°$ calibration method (Freudenthaler, 2016) three times a

day, therefore calibration factors were readily available. To correlate the vertically retrieved lidar optical properties to surface observations, we computed the mean lidar aerosol optical properties of the first detected layer. The geometrical boundaries of the first layer expanded from the surface all the way up to the top of the Boundary Layer (BL). In order to detect the upper boundary of the BL, the derivative of the range-corrected signal at 1064 nm was considered (Menut et al., 1999). The inflection point (second derivative is zero) indicated the BL top. Within this layer the variation of optical properties may largely fluctuate.

This is subject to the aerosol amount, type, distribution, as well as the height of the BL. In order to reduce the uncertainty introduced by these factors at the top of the geometrical layer, we have removed height bins which their value was lower than 25% from the mean value of the layer. This feature was mainly present at the shorter wavelengths towards the top of the BL in which the optical properties were effectively changing.

Similar procedure was followed for the retrieval of the optical profiles from CL61 ceilometer. The forward Klett inversion
(Wiegner and Gasteiger, 2015) was applied to the 2h temporally averaged signals using a constant LR of 60 sr. In this way, the retrieval is possible under low cloud conditions broadening the applicability of the lidar to retrieve aerosol information within the BL. The stratocumulus cloud technique was used to determine the value of the calibration factor and its inter-annual variability (O'Connor et al., 2004; Hopkin et al., 2019). The Finnish Meteorological Institute operates five CL61 units and tests have shown that the factory calibration is within 15 % accuracy. Two out of five CL61 instruments located above 65° N

latitude exhibit an inter-annual discrepancy of 3-5 % from the factory calibration. Further discrepancies in the southern stations, including the one in this study, may emerge from the water vapor amount or/and a range of possible multiple-scattering factors bounded to the selection of the cloud cases themselves. Therefore, both a water vapor correction and an adequate calibration correction applied to the signals. For the water vapor correction, the methodology described in Wiegner and Gasteiger (2015) study was applied. Regarding the calculation of the PDR, the theoretical molecular depolarization value of 0.0036 was utilized

(Behrendt and Nakamura, 2002). As a reference, a 100 % departure from the theoretical value (Burton et al., 2015) introduced an increase of 3 % in the PDR. This uncertainty was included on top of the reported PDR variability for birch and pine. Profiles below 200 m above ground level were considered constant to the value at 200 m.

Lastly, vertically-pointing data from Halo Doppler lidar were post-processed following Vakkari et al. (2019). Similar to previous studies on aerosol PDR using Halo Doppler lidars, bleed-through of the internal polarizer was estimated from liquid

cloud base observations as $0.011 \pm 0.005$, which is very close to the previous values of 0.016 and 0.013 for this system (Vakkari et al., 2021; Bohlmann et al., 2019). PDR at 1565 nm was corrected for the bleed-through following Vakkari et al. (2021). Also, attenuated backscatter at 1565 nm was calculated from the post-processed, 2 h averaged co-polar signal-to-noise-ratio following Vakkari et al. (2021). Profiles below 100 m above ground level were considered constant.



Using the automatically detected BL from PollyXT, the layer mean intensive optical properties of BÅE and PDR from all three lidars were calculated and further connected to surface pollen and $PM_{10}$ concentrations. For each pollen type, cases with a greater or equal than 90 % of share of that pollen compared to the rest pollen types were considered to assure that the effect of that specific pollen alone is studied.

## 3 Results and discussion

Year 2021 was exceptional in terms of airborne pollen concentration in many areas of Finland (European Aeroallergen Network (EAN) pollen database). At Kuopio pollen monitoring site (62°8′ N, 27°63′ E; 98 m above sea level) which is in close proximity to Vehmasmäki station and has an extended pollen monitoring dataset, the total accumulated birch pollen reached the second-highest count of the continuous 43-year-long history of airborne pollen monitoring. The yearly variation in airborne pine pollen concentration on this site is less pronounced but during 2021 pine concentration was higher than average. Pollen monitoring at Vehmasmäki station has been available since 2016. Figure 1 shows the pollen concentration of birch and pine for the two consecutive years of 2021 and 2022 at Vehmasmäki station. It can be observed that the pollination season and pollen amount fluctuate from year to year. Specifically, intense pollination started a week later in 2022 compared to 2021 regardless the pollen type. The diurnal variation of birch and pine is also visible in which pollen concentration normally peaks during local noon following the growth of BL height. The maximum pollen concentration for birch and pine ever recorded on site is 64380 and 14550 particles $m^{-3}$, respectively. For birch this maximum concentration occurred during 2021 while for pine, the highest amount of 10570 particles $m^{-3}$ in 2021 is relatively close to the all-time maximum. As a reference, considering the combined Kuopio-Vehmasmäki pollen dataset, 95 % of the time pollen concentration of less than 6000 and 3500 particles $m^{-3}$ is observed on site for birch and pine, respectively.

### 3.1 Pollen optical properties

The PM concentration and pollen types were linked to layer optical properties considering the first atmospheric layer as retrieved from the lidar observations (see Sect.2.6). The top height of the first layer ranged between 660 and 2900 m and it was subject to the evolution of the BL. For each pollen type, cases with a greater or equal than 90 % of share of that pollen compared to the rest pollen types were considered to assure that the effect of that specific pollen alone is studied.

*Birch*

An example case of the methodology is shown in Figure 2 during the high birch pollination on 12th of May 2021. Figure 2 illustrates the concentration of pollen summarized as birch and other pollen types measured at 4 m above ground level by the Hirst-type air sampler (Fig 2a), the range-corrected signal at 1064 nm (Fig 2b) and the intensive and extensive lidar optical properties (Fig 2c-e) considering time-averaged profiles from 11 to 13 UTC (shaded area in Fig 2a). Supporting information regarding the presence of smoke and mineral dust particles are summarized in Figure 3. During the day, pollen concentration remained at high levels and birch pollen was the dominant type with little to no presence of other pollen types. Regarding





the presence of other aerosol types, BC concentration increased towards the end of the day indicating the presence of smoke particles. Minor presence of mineral dust particles was also hinted by some of the dust models towards the second half of the day. The top of the BL has reached as high as 1.65 km during the day, showing a clear signature in the particle backscatter profiles as well (Fig 2c). A LR of 55 (60 sr) at 355 (532, 910 and 1064 nm) was used for the Klett inversion, respectively.

Regardless of the wavelength, the particle backscatter coefficient showed minimal discrepancy within the BL suggesting well mixed conditions. Towards the top of the boundary layer, the particle backscatter coefficient at 532 nm was larger than at 355 nm, resulting to slightly negative $\text{BÅE}_{355/532}$, a feature that has previously seen in the presence of mineral dust aerosol particles (Veselovskii et al., 2016). Considering the data in the grey shaded area (first geometrical layer), optical layer mean values of $0.15 \pm 0.23$, $0.19 \pm 0.04$ and $0.51 \pm 0.01$ were calculated for the $\text{BÅE}_{355-532}$, $\text{BÅE}_{532-910}$ and, $\text{BÅE}_{532-1064}$,

respectively. Furthermore, little discrepancy can be seen at the PDR profiles at the two longer wavelengths of 910 and 1565 nm. This is not the case with the $\text{PDR}_{532}$ profile in which the value decreases with increasing height. It is presumed that the concentration of pollen decreases with height thus, the influence of background aerosol increases. Since the longer wavelengths of 910 nm and 1565 nm are more sensitive to large aerosol particles the depolarization ratio does not decrease with height. The opposite is valid for the 355 and 532 nm wavelengths which are more sensitive to small aerosol particles. Because of this, the

$\text{PDR}_{532}$ values in the upper end of the signal considerably decreased and therefore they are not considered for the calculation of the layer mean values (see Sect.2.6). A layer mean value of $4 \pm 1\%$, $29 \pm 4\%$, $24 \pm 1\%$ and $36 \pm 1\%$ was retrieved for the PDR at 355, 532, 910 and 1565 nm, respectively.

A total of 102 cases (individual layers) were considered during the years of 2021 and 2022 for the exploration of the pollen optical properties. Figure 4 demonstrates the layer mean PDR in relation to the birch pollen concentration for all available

wavelengths. The color indicates the share of $\text{PM}_{10}$ to $\text{PM}_{10}$ plus $\text{PM}_{birch}$. The size of the marker shows the mean relative humidity (RH) at 26 m above ground level. The RH fluctuated between 22% and 95 %, covering a considerable range of values. At 26 m above ground level, wind gusts up to 13 m s$^{-1}$ prevailed. Considering all cases, a mean $\text{PM}_{10}$ concentration of $15 \pm 13$ $\mu$g m$^{-3}$ (range: 2-78 $\mu$g m$^{-3}$) was obtained. Focusing on the two longest wavelengths, it is evident that the pollen concentration (contribution of other aerosols) is positively (negatively) correlated to the PDR. The higher the pollen

concentration and the lower the contribution of other aerosols resulted to higher PDR in the pollen layer. There were a few points in the high birch concentration region (more than 20000 particles m$^{-3}$) with relatively lower PDR values. These cases mainly occurred during the 12th of May 2021 (at 08 UTC and from 20 to 22 UTC) and 13th of May 2021 (from 04 to 18 UTC) as opposed to the higher PDR cases which they were observed during the night of 11th of May 2021 (from 18 to 22 UTC) and 12th of May 2021 (at 06 UTC and from 10 to 18 UTC). The low PDR cases featured higher share of $\text{PM}_{10}$ with

a mean $\text{PM}_{10}$ concentration of $38 \pm 20$ $\mu$g m$^{-3}$ compared to $18 \pm 4$ $\mu$g m$^{-3}$ at higher PDR cases, and slightly higher mean RH values ($40 \pm 13$ % as opposed to $36 \pm 12$ %). To support further the analysis, Figure 3 demonstrates that during the low PDR times the BC concentration was up to 3 times higher compared to other times in which the BC concentration was around 0.1 $\mu$g m$^{-3}$, marking the presence of smoke aerosol particles from forest fires. At the same time, modeled dust optical depth hinted to long-range transported mineral dust. Therefore, the lower PDR values at the high concentration region stemmed from

a mixture of pollen, dust and smoke aerosol particles.





In order to retrieve the characteristic optical properties of birch pollen, cases with the lowest $PM_{10}$ contribution (less than 10%) to the aerosol mixture was considered. The 10% limit was a compromise of having meaningful number of cases for the calculation of the birch pollen optical properties. A mean (range of) birch PDR of $5 \pm 2\%$ (4-8) , $28 \pm 6\%$ (21-35), $23 \pm 6\%$ (17-26) and, $33 \pm 4\%$ (27-38) at 355, 532, 910 and 1565 nm wavelengths was estimated, respectively. This subset of 7 cases

had a $BÅE_{355-532}$, $BÅE_{532-910}$ and, $BÅE_{532-1064}$ of $0.37 \pm 0.63$, $0.16 \pm 0.20$ and $0.54 \pm 0.41$, respectively. In this subset, the share of birch pollen compared to rest of pollen types ranged between 99.5% and 100% and no pine or spruce pollen were present. The values indicate rather large aerosol particles and they are aligned with the hypothesis used in Shang et al. (2020) for pollen. Figure 5 summarizes the spectral dependence of birch PDR from this study and literature values. Although most of the studies have been conducted in a controlled laboratory environment using purchased pollen grains, we concluded to a

similar wavelength dependence using naturally-occurred pollen in ambient conditions. The related findings are comparable as the total of 7 cases considered for the retrieval of birch PDR were found in dry conditions scoring a RH of $29 \pm 9$ %. This range of RH is similar to the ones reported in the laboratory studies. Nevertheless, whether the PDR values refer to intact pollen grains, ruptured pollen or fragments of them, it is unclear. Ruptured pollen has an altered shape compared to the intact and hydrated pollen grain. Furthermore, smaller fragments of ruptured pollen are also non-spherical, therefore potentially affect-

ing the PDR. Pollen fragments are usually smaller than 2.5 $\mu$m (Mampage et al., 2022). Hence, this effect is expected to be more profound at the smaller wavelengths which are most sensitive to smaller aerosol particles than the longer ones. From the pollen characterization perspective, there is a sharp increase from 355 nm to 532 nm wavelength. Further, PDR values decrease slightly at 910 nm before increasing again towards the longest wavelength of 1565 nm. As demonstrated through Figure 4 the discrepancy in the birch $PDR_{532}$ and $PDR_{532}$ values between this study and the studies of Bohlmann et al. (2019, 2021),

and Shang et al. (2022) can be attributed to the consideration of the amount, hydration and share of pollen and other aerosol particles in the mixture.

*Pine*

Likewise, Figure 6 presents equivalent cases for pine pollen. In total, 110 individual layers were considered. The RH ranged

between 25 % and 97 % and wind gusts up to 9.5 m s$^{-1}$ were measured at 26 m above ground level. Considering all cases, a mean $PM_{10}$ concentration of $12 \pm 6$ $\mu$g m$^{-3}$ (range: 4-42 $\mu$g m$^{-3}$) was obtained. Regarding the PDR, higher mean layer values were apparent for the wavelengths of 355 nm and 532 nm compared to birch pollen while the $PDR_{1565}$ presented lower values compared to birch pollen. The $PDR_{910}$ had comparable values to birch. It can be observed that the longer wavelengths reached a plateau above certain pine pollen concentration. At the same time, there was a wide discrepancy of values at $PDR_{532}$ which

almost monotonically increased with increasing pine concentration. The $PDR_{532}$ peaked during high pine concentration cases resulting to values as high as 66 %. Previously, Bohlmann et al. (2021) have reported maximum $PDR_{532}$ as high as 90% in the presence of spruce pollen, a similar pollen type in terms of shape to that of pine pollen. This feature has been also detected here for pine pollen. The presence of other depolarizing aerosols is excluded. Furthermore, no multiple scattering has been found for the preceding instruments at the highest observed pollen concentration. The multiple scattering calculations performed using

Eloranta (1998) model. To investigate the overall fluctuation at $PDR_{532}$, we regarded the aerosol volume size distribution of



the particles considering cases with pine concentration above 4000 particles m$^{-3}$. There were 13 cases which they were further split into three categories (PDR$_{532} \leq 40\%$ (2 cases), $40 <$ PDR$_{532} < 60\%$ (8 cases) and PDR$_{532} \geq 60\%$ (3 cases)). In terms of RH there was no significant difference between the three categories with mean layer values of 43% (low PDR$_{532}$), 40% (mid PDR$_{532}$) and 34% (high PDR$_{532}$), respectively. Similarly, the mean PM$_{10}$ concentration was 11, 13, and 12 $\mu$g m$^{-3}$ for the low, mid and high PDR$_{532}$ categories, respectively. Therefore, the discrepancy in the PDR$_{532}$ was attributed to the relative contribution of the different aerosol modes in the aerosol volume size distribution (Fig. 7). Considering this dataset, pollen optical properties are dominated by large particles ($> 1$ $\mu$m). Specifically, between the high and low PDR$_{532}$ categories there was a noticeable change in the share of particles with diameter between 1 and 2.5 $\mu$m. Particularly, a 21% share was attributed to this size range for the low PDR$_{532}$ category as opposed to the high PDR$_{532}$ category in which the share was as low as 6%. Whilst, particles with size below 1 $\mu$m had a 5-11% bigger share in the high-mid PDR category compared to the low PDR category. The mid PDR category presented almost identical aerosol size distribution to the high PDR category, thus the change in the PDR$_{532}$ value may have resulted from changes in the structure of the pollen itself. At the same concentration range, the longer wavelengths showed little PDR discrepancy. Consequently, the longer wavelengths can be beneficial for the detection of the large pollen particles.

Considering cases with PM$_{10}$ contribution of less than 5% in the total aerosol share, a mean PDR of $6 \pm 2\%$, $43 \pm 11\%$, $22 \pm 6\%$ and, $26 \pm 3\%$ at 355, 532, 910 and 1565 nm was retrieved, respectively. The mean RH for this subset of 30 cases was $42 \pm 9\%$. Similar to Figure 5, Figure 8 presents the spectral dependence of PDR for pine pollen. There is a maximum observed at 532 nm wavelength for this pollen type. This is evident at all studies (laboratory and ambient ones). We found a minor spectral dependence between 910 and 1565 nm wavelengths compared to Cao et al. (2010) in which the PDR decreases with increasing wavelength. Regarding the PDR$_{355}$, the combination of the large pine particles which may not be distributed homogeneously in the BL and the instrumental limitation of no reliable measurements below 800 m resulted to uncorrelated PDR$_{355}$ values compared to the laboratory studies. Noteworthy is the similarity of the spruce PDR spectral dependence in Bohlmann et al. (2021) and this study for pine pollen. Although the layer mean BÅE was $0.14 \pm 0.49$, $0.53 \pm 0.42$ and, $0.47 \pm 0.24$ at BÅE$_{355-532}$, BÅE$_{532-910}$ and, BÅE$_{532-1064}$, respectively, negative values were retrieved for the BÅE$_{355-532}$ during the highest pine pollen cases. Negative BÅE has been reported previously by Bohlmann et al. (2021) during high concentrations of spruce pollen. To the same direction, negative BÅE values have been previously retrieved in the presence of mineral dust aerosol particles. Recently, Veselovskii et al. (2020) have linked the discrepancy of BÅE values in dust layers to the complex refractive index of the particles and more specifically to the imaginary part of the complex refractive index. To this end, the imaginary part of the complex refractive index for pine pollen remains undocumented, yet the negative values at BÅE$_{355-532}$ combined with the wavelength dependence and the maxima at PDR$_{532}$ can be a general characteristic of pollen belonging to the family of *Pinaceae*.



## 4   Summary and conclusions

Pollen optical properties of birch and pine were investigated with a synergy of three lidars and collocated in situ aerosol and

pollen instruments in the rural site of Vehmasmäki (Kuopio) in Finland. Regardless of the pollen type, the PDR was positively (negatively) correlated with the pollen concentration (share of other aerosol particles in the mixture). The higher the pollen concentration and the lower the contribution of other aerosols resulted to higher PDR in the pollen layer. More specifically, the PDR at both 910 and 1565 nm wavelengths reached a plateau at high pollen concentration with minimal discrepancy. The reason behind is the sensitivity of the longer wavelengths to large particles like pollen. This marks the suitability of the PDR at

longer wavelengths for pollen detection. Being most sensitive to small particles, the shorter wavelengths of 355 nm and 532 nm demonstrated a larger PDR variability even at high pollen counts. Whether this PDR discrepancy at shorter wavelengths and specifically at 532 nm is attributed to the share of pollen (intact, ruptured or fragments) in the mixture or it is the result of more complex mechanisms, for example pollen shape and orientation, it is not feasible with the current instrumental set up. Therefore, the findings refer to the effect of pollen (intact, ruptured or fragments) in the aerosol mixture.

By choosing cases with low background aerosol population, we were able to retrieve the characteristic PDR and BÅE optical properties for birch and pine pollen. Regarding birch pollen, we retrieved a PDR of $5 \pm 2\%$, $28 \pm 6\%$, $23 \pm 6\%$ and, $33 \pm 4\%$ at 355, 532, 910 and 1565 nm wavelengths, respectively. In terms of BÅE, birch pollen had similar values to dust aerosols indicating rather big particles. A mean value of $0.37 \pm 0.63$, $0.16 \pm 0.20$ and $0.54 \pm 0.41$ was calculated for $BÅE_{355-532}$, $BÅE_{532-910}$ and, $BÅE_{532-1064}$ combinations at the pollen layer, respectively. Birch pollen layers were found in

rather dry conditions with a mean RH of $29 \pm 9\%$. Regarding pine pollen, even higher PDR values were estimated for 532 nm wavelength compared to that of birch. The pollen layers had a mean PDR of $6 \pm 2\%$, $43 \pm 11\%$, $22 \pm 6\%$ and, $26 \pm 3\%$ at 355, 532, 910 and 1565 nm wavelengths, respectively. As far as BÅE concerns, values of $0.14 \pm 0.49$, $0.53 \pm 0.42$ and, $0.47 \pm 0.24$ were found at $BÅE_{355-532}$, $BÅE_{532-910}$ and, $BÅE_{532-1064}$, respectively. Negative $BÅE_{355-532}$ were retrieved during the highest pine pollen concentrations. The negative BÅE together with the high PDR at 532 nm could be characteristic

for this pollen type.

Pollen release in the Northern Hemisphere is well-documented yet little is known about its optical properties. Lidar-derived PDR and BÅE profiles can provide essential information about pollen presence in the atmosphere serving as a validation tool in aerosol dispersion models of pollen and further improving pollen representation through lidar assimilation procedures. Furthermore, pollen is currently missing from lidar aerosol classification algorithms. To this end, there is a great inconsistency in

the lidar literature related to the characteristic optical properties of atmospheric pollen. Here, we have hinted that discrepancies in the background aerosol population and the pollen amount with regard to the wavelength and possibly the pollen size itself are most probably the reasons behind this discrepancy. The challenge of defining pollen optical properties lay on their low concentration in the atmosphere. Because of the relatively clean background conditions on our measurement site, we were able to capture the relationship between PDR and pollen concentration and further connect it to the presence of other aerosol

particles in the air. Nevertheless, with the current instrumental set up we were not able to distinguish whether the reported



optical properties are the result of intact pollen grains, ruptured pollen, fragments or a combination of these. Therefore, future studies should focus on characterizing the state of the pollen grains and further link these to optical properties.

*Data availability.* Level 1 PollyXT observations are available at https://polly.tropos.de/ (last access: 20 March 2023). L1 data for the CL61 ceilometer and the Halo Doppler lidar are available through CLOUDNET portal at https://cloudnet.fmi.fi/ (last access: 20 March 2023).

Level 2 lidar data and in situ data are available upon request from the authors. The GDAS meteorological data used in Klett inversion are available at https://www.ready.noaa.gov/HYSPLIT.php (last access: 20 March 2023).

*Author contributions.* MF developed the algorithms and analysed the PollyXT and CL61 lidar data, conceptualized and wrote the original paper. AL developed the algorithms and analyzed the in situ aerosol data. VV developed the algorithms and provided the HALO Doppler data. AS and LT performed the analysis of the pollen samples. EOC performed the multiple scattering calculations. AL, PT, XS and MK

were responsible for the lidar and in situ good operation and the collection of the pollen samples during the campaigns. MT, PT and EOC assisted with the CL61 calibration and data interpretation. All authors were involved in editing the paper, interpreting the results, and the discussion of the manuscript.

*Competing interests.* No competing interests are present.

*Acknowledgements.* The authors gratefully acknowledge the support of Vaikuttavuussäätiö (Finnish Research Impact Foundation) through

the Tandem Industry Academia (TIA) program. Dust data and/or images were provided by the WMO Barcelona Dust Regional Center and the partners of the Sand and Dust Storm Warning Advisory and Assessment System (SDS-WAS) for Northern Africa, the Middle East and Europe.



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



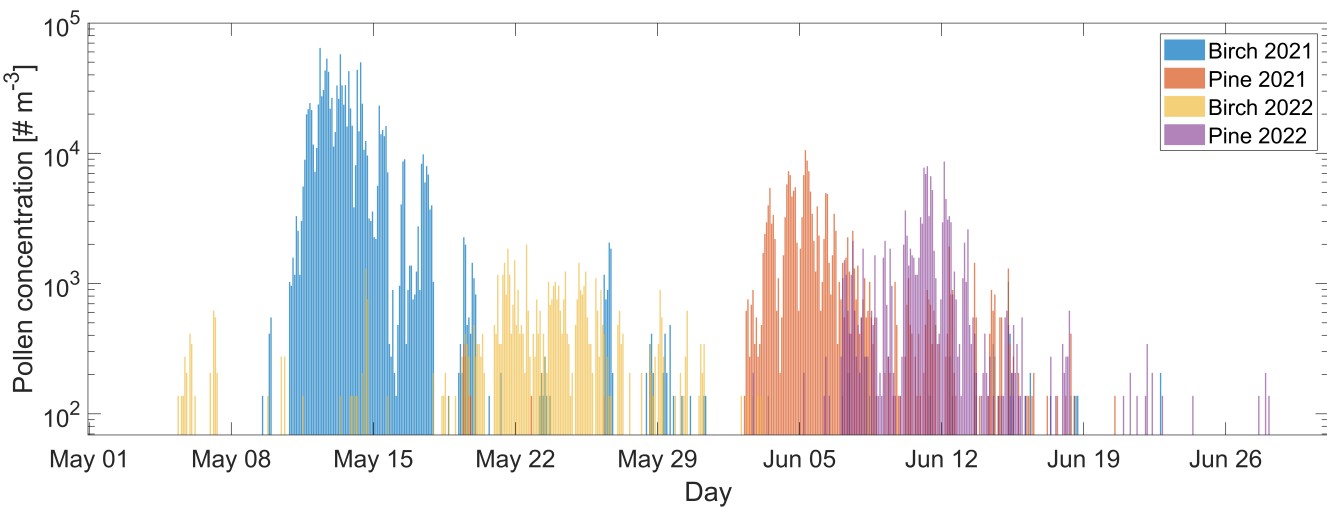

**Figure 1.** Timeseries of birch and pine pollen concentration measured by the Hirst-type air sampler at 4 m above ground level for the years of 2021 and 2022 at Vehmasmäki station in Finland



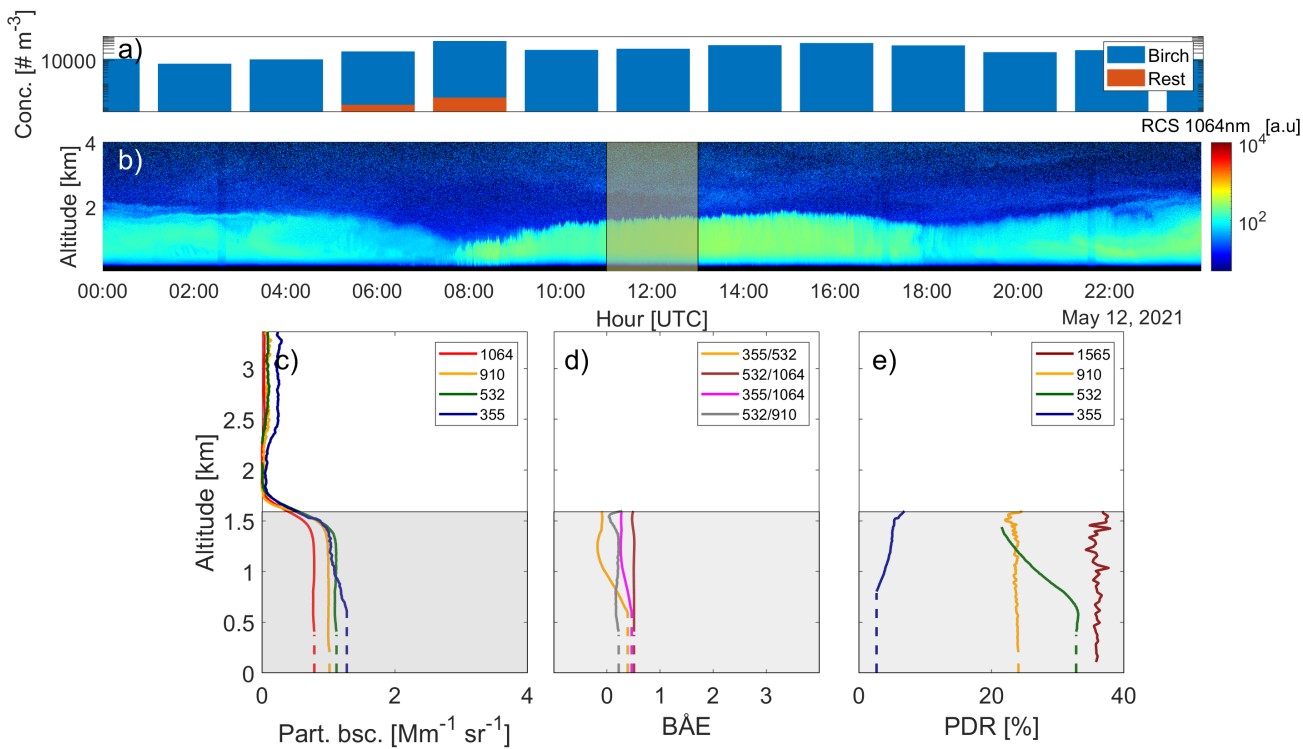

**Figure 2.** Overview of the 12th of May 2021. a) Pollen concentration measured by the Hirst-type air sampler at 4 m above ground level. b) Range-corrected signal (RCS) at 1064 nm. The shaded area indicates the time range considered for the Klett inversion shown below. Profiles of c) particle backscatter coefficients at 355, 532, 910 and 1064 nm, d) BÅE at various combinations and e) PDR at 355, 532, 910 and 1565 nm are shown, respectively. The first geometrical layer is marked with grey colour.



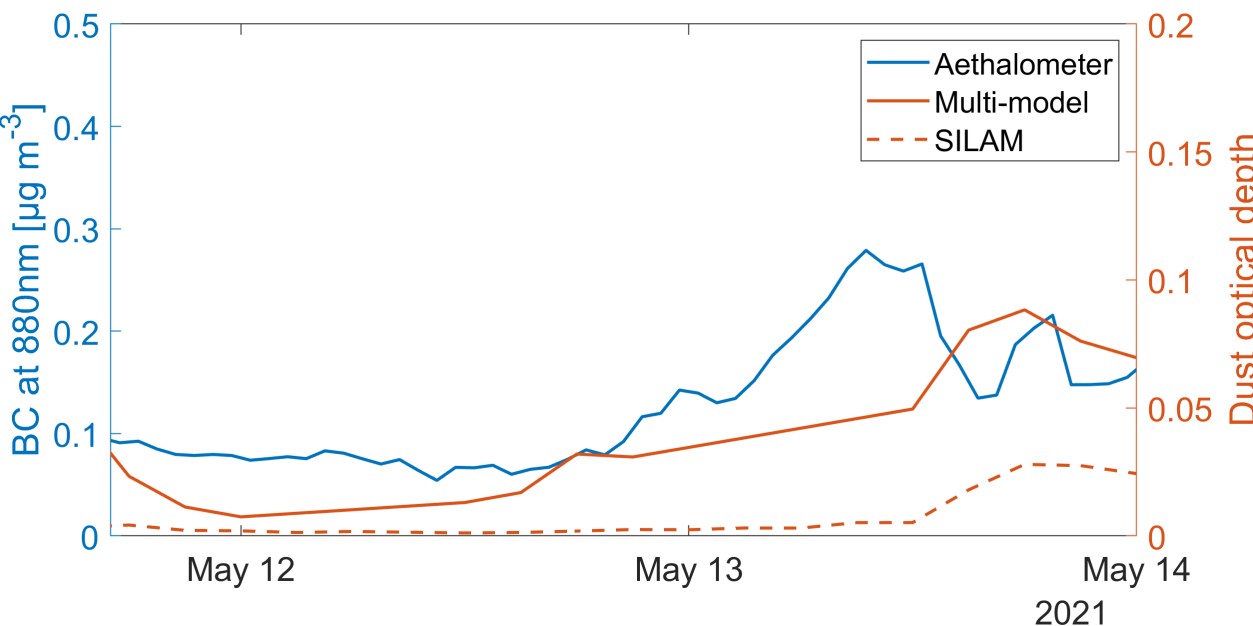

**Figure 3.** Left: On site aerosol black carbon (BC) concentration from the AE31 Aethalometer between 11th and 13th of May 2021. The 880 nm wavelength was used for the BC concentration. Right: Corresponding modeled dust optical depth provided by the WMO Barcelona Dust Regional Center (multi-model and SILAM options are shown for comparison).

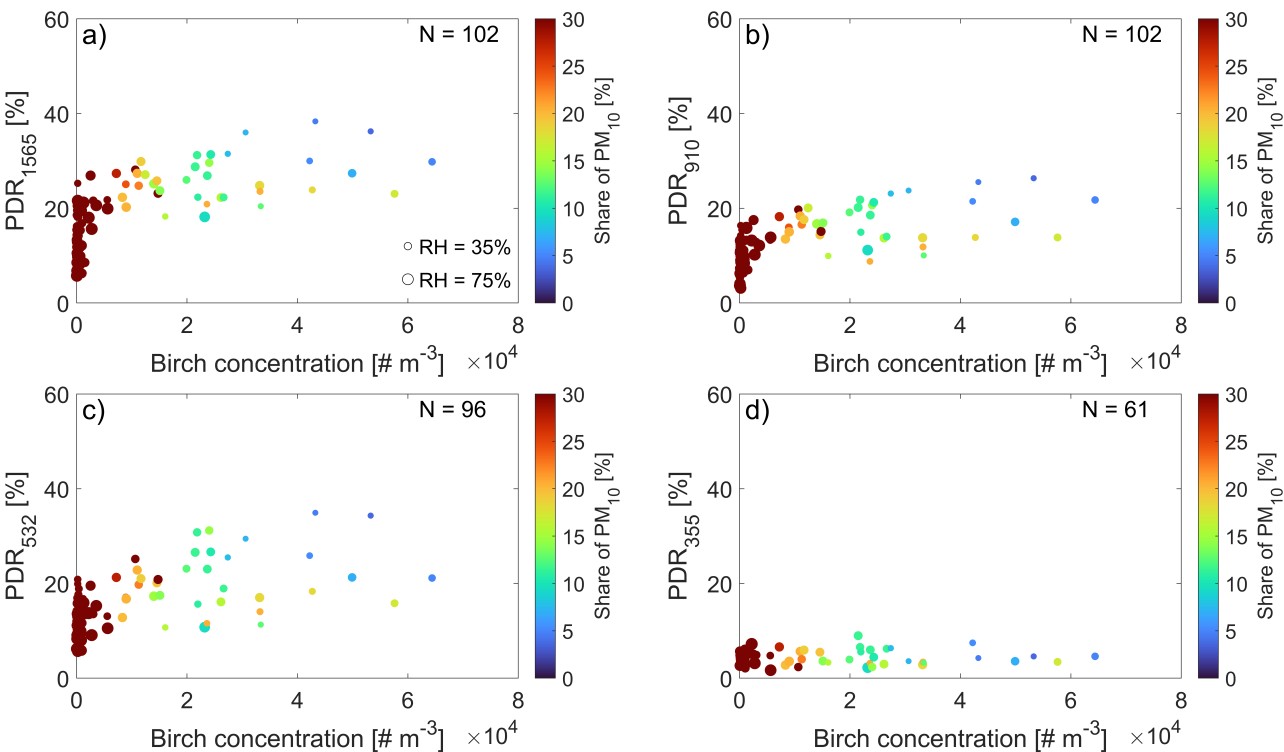

**Figure 4.** Wavelength dependence of the PDR at a) 1565 nm b) 910 nm c) 532 nm and d) 355 nm on the birch pollen concentration. The size of the circles indicate the RH at 26 m above ground level. The color of the circles correspond to the share of $PM_{10}$ in the $PM_{10}$ plus $PM_{birch}$ sum. The number at the top right corner corresponds to the number of layers considered at each wavelength.





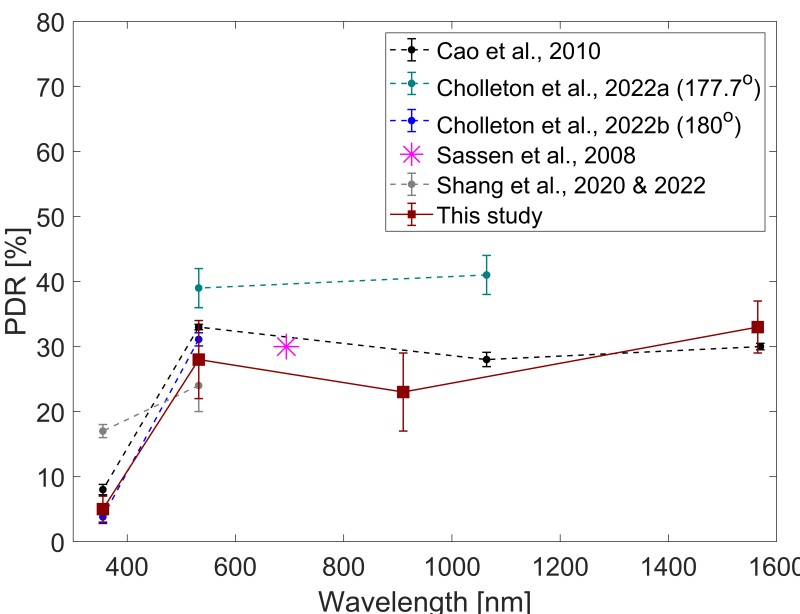

**Figure 5.** Spectral dependence of the birch pollen PDR. Error bars represent the standard deviation. Note: Sassen et al., (2008) and Cao et al., (2010) studies refer to *Betula papyrifera*. Cholleton et al. studies realized at two different back-scattering angles. Shang et al. studies refer to modelled birch (*Betula*) assuming a BÅE of 0.

.



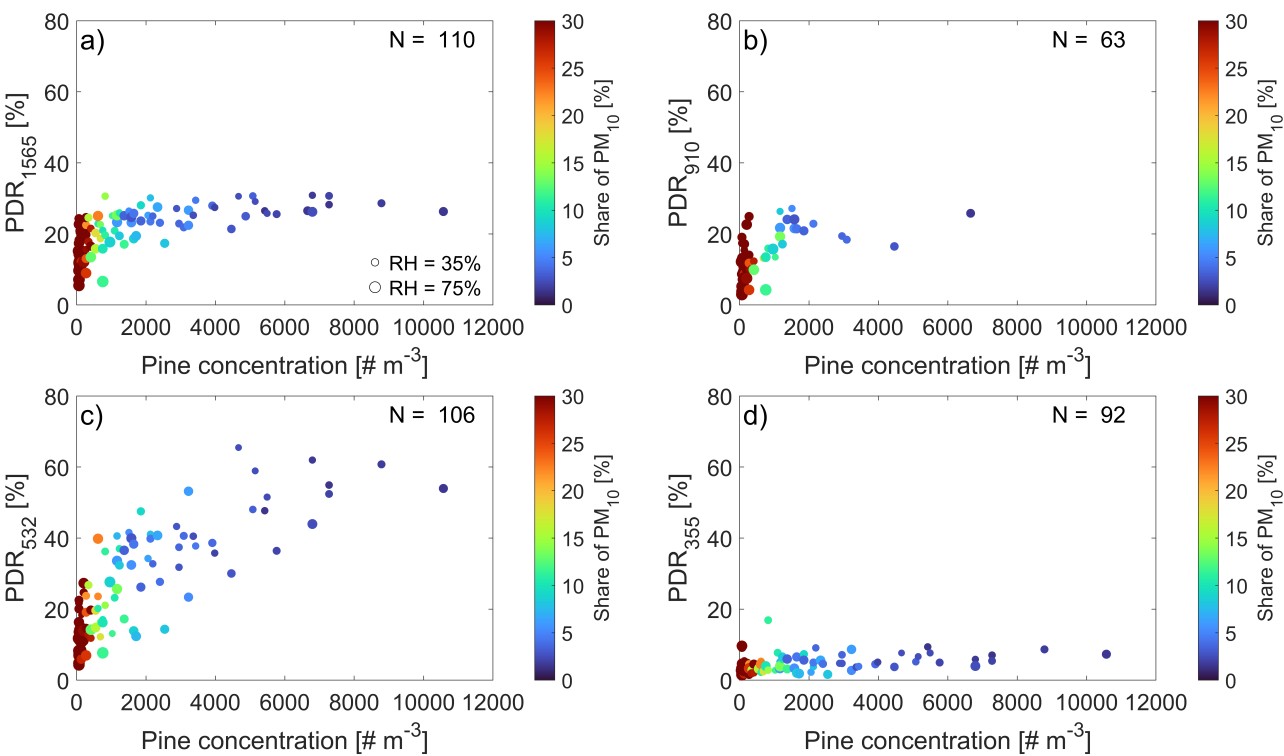

**Figure 6.** Wavelength dependence of the PDR at a) 1565 nm b) 910 nm c) 532 nm and d) 355 nm on the pine pollen concentration. The size of the circles indicate the RH at 26 m above ground level. The color of the circles correspond to the share of $PM_{10}$ in the $PM_{10}$ plus $PM_{pine}$ sum. The number at the top right corner corresponds to the number of layers considered at each wavelength.



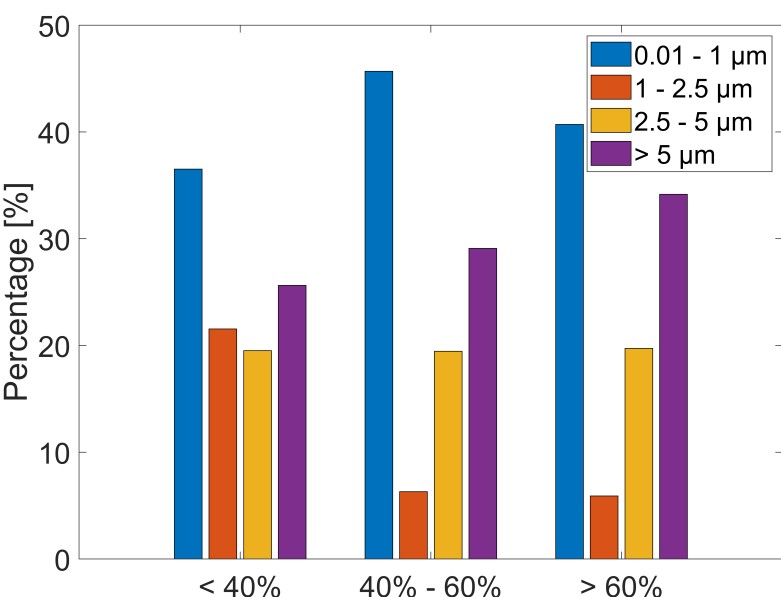

**Figure 7.** Relative contribution of PM in the size range of 0.01 - 1 $\mu$m (blue), 1 - 2.5 $\mu$m (red), 2.5- 5 $\mu$m (yellow) and > 5 $\mu$m (purple) to PM$_{10}$ as detected by NS and OPS instruments. The data shown correspond to cases with PDR$_{532}$ larger than 60%, between 40% and 60% and below 40%. All three categories were limited to pine pollen concentration above 4000 particles m$^{-3}$.

.



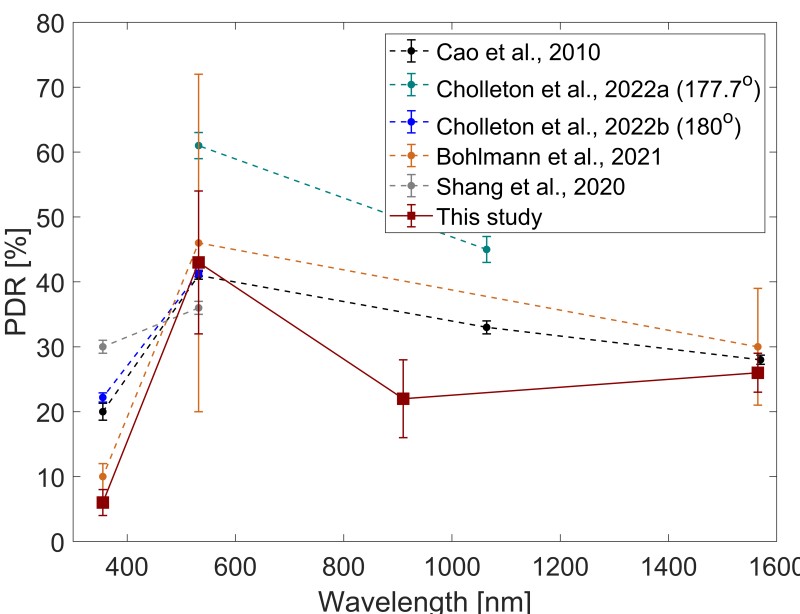

**Figure 8.** Similar to Figure 5 for pine pollen. Cao et al., (2010) study refers to *Pinus virginiana* and Cholleton et al. studies refer to *Pinus strobus*. Bohlmann et al, (2021) study refers to a mixture of *Betula* and Norway spruce (*Picea abies*). Shang et al. (2020) study refers to modelled pine (*Pinus*) assuming a BÅE of 0.

.