# Peer review of "Spectral dependence of birch and pine pollen optical properties using a synergy of lidar instruments"

_EGUsphere, 2023_

## Author Comment (AC1)

*The authors would like to thank the reviewer for his/her valuable comments and suggestions. We have modified the manuscript with the proposed changes along with step-by-step answers to the suggestions. Please note that changes have been highlighted (in bold or 'track changes') in the manuscript and the corresponding answers to the reviewer by text below. The original comments are presented in bold letters.*

Reviewer #3

**The authors present both active remote sensing measurements and in situ aerosol observations of birch and pine pollen particles at Vehmasmäki (Kuopio), Finland to analyze the spectral dependence of the particle depolarization ratio (PDR). A special feature of this study is that the concurrent use of three different lidar systems enables measurements of PDR at four different wavelengths. Furthermore, the measurements were conducted at a rural forest site, with typically reduced pollution in the background. This is of great value as it allows the authors to study optical properties of pure pollen which is especially important as pollen have typically low concentrations compared to other atmospheric aerosol types.**

**Therefore, this study is important and suitable for ACP. Overall, the paper is well written and of good quality. Only some few aspects could be explained further/expressed clearer and several typos should be corrected, which will be addressed in the special comments below.**

**Special comments:**

**ll. 60-61: Veselovskii et al. (2021) only used a single broadband fluorescence channel in this study. An approach to obtain spectral fluorescence information by several fluorescence channels has just been presented in Veselovskii et al. (2023, preprint)**.

Thank you for your comment. We have included the suggested study in the manuscript.

**ll. 136-137: I don't understand what is meant with the statement that NS and OPS aerosol size distributions were combined, but NS size distribution was neglected. If only the OPS size distribution was used, then it wasn't combined with the NS one, was it? Could you please clarify this**?

Indeed, the sentence is incomplete and misleading. Only the last two bins of the NS size distribution were neglected, not the whole size distribution. We have corrected the sentence.

**ll. 206-207: The statement 'For each pollen type … of that specific pollen alone is studied' has already been stated in the same way in ll. 185-187 and is thus repetitive and could be removed.**

Removed as suggested.

**ll. 238-240, Fig. 4: Also, the 532 nm PDR seems similarly correlated to pollen concentration and concentration of other aerosols as the two longest wavelengths. Only the PDR at 355 nm seems less influenced. Why? Do you also relate this to the higher sensitivity of longer wavelengths to the comparably large pollen particles? Please explain your conclusion further here.**

The reviewer raises a valid comment. One important factor to be considered here is the pollen distribution within the PBL and its relative share to other aerosols. Birch is a big particle (~25μm) and at the same time smaller compared to pine pollen (~75μm) and its distribution may change with height. Given the scenario that we have a well-mixed PBL with constant share of background aerosols and birch pollen then the height limitation of the PDR at 355nm should not be a problem, depicting the birch optical properties. In other times, the 800m height can be limiting enough to see the full effect of birch on the $PDR_{355}$ when the PBL is not well-mixed or if the concentration of birch is not homogeneously distributed within PBL (well-mixed or not). In this context, the longer wavelengths do present an advantage for pollen detection as they are less influenced by the concentration/presence of smaller particles. This study contains optical properties for a wide range of birch concentrations. In fact, the higher end of birch concentrations occurred at the measurement site are exceptional and not frequent at all. Therefore, we conclude that the birch PDR at 355nm wavelength have been sensed adequately in ambient conditions. This conclusion is backed up by laboratory studies as well which found similar behaviour at 355nm compared to longer wavelengths.

**Figure 6: Why do you have less PDR values at 910 nm (only 2?) for pine concentrations > 4000 m$^{-3}$ compared to the other wavelengths? Was there a technical issue with the respective lidar? Maybe you could add a short remark on that, please.**

The reviewer is correct. The CL61 ceilometer was not operational for a few days during the pine season in 2021 which resulted to fewer cases at this specific wavelength. More specifically, out of the 110 pine cases, 77 occurred during 2021 and 33 cases during 2022. Although both years had comparable pine concentrations, the low cloud conditions during most of the high pine concentration in 2022 limited the derivation of the optical properties. We have added the following sentence to the manuscript:

'Note that the CL61 instrument was not operational the full pine period and thus has lower number of cases.'

**Typos and technical notes:**

**l. 19: '… is closely associated to allergic diseases' --> is closely associated with allergic diseases**.

Corrected as suggested.

**l. 28: 'SSPs' --> 'SPPs'**

Corrected.

**l. 31: 'are an effective ice nuclei' --> are effective ice nuclei (as it is plural)**

Noted and corrected as suggested.

**l. 45: 'depolarisation' --> depolarization**

Corrected.

**l. 60: 'polarisation' --> polarization**

Corrected.

**l. 82: 'vary from year to another' --> vary from one year to another**

Corrected as suggested.

**l. 103: 'can be found at Vakkari et al. (2021)' --> can be found in Vakkari et al. (2021)**

Corrected.

**l. 205: 'Sect.2.6' --> Sect. 2.6 (missing space)**

Corrected.

**l. 212, l. 213: don't forget the point after Fig. …, e. g., 'Fig 2a' --> Fig. 2a**

Thank you for your comment. We went through the manuscript, and we have added the missing dots.

**l. 213: 'Fig 2a)' --> Fig. 2b) (the shaded area is found in panel b)**

Corrected.

**l. 222: 'has previously seen' --> has previously been seen**

Corrected.

**l. 284: 'calculations performed' --> calculations were performed**

Corrected.

**l. 328: 'set up' --> setup (as it is the noun)**

Noted and corrected.

**References:**

Veselovskii, I., Kasianik, N., Korenskii, M., Hu, Q., Goloub, P., Podvin, T., and Liu, D.: Multiwavelength fluorescence lidar observations of fresh smoke plumes, Atmos. Meas. Tech. Discuss. [preprint], https://doi.org/10.5194/amt-2023-5, in review, 2023.

---

## Author Comment (AC2)

*The authors would like to thank the reviewer for his/her valuable comments and suggestions. We have modified the manuscript with the proposed changes along with step-by-step answers to the suggestions. Please note that changes have been highlighted (in bold or 'track changes') in the manuscript and the corresponding answers to the reviewer by text below. The original comments are presented in bold letters.*

Reviewer #2

**The main research content of this paper is the difference in depolarization ratio by wavelength depending on the type of pollen. It is judged to have important value as a paper with high continuity with previously published papers related to pollen. Overall, it is judged to be of excellent quality, but it is judged that it needs some revision.**

**Please refer to the information below.**

**Since the PDR value of pollen changes depending on mixing with aerosols other than pollen, such as PM10, it would be good to include this information in the text. In the current thesis, it is indicated in the graph, but it is not separately indicated in the text. It would be better to distinguish the PDR value when it is pure pollen and the value when mixed and indicate the average value in the text**.

Thank you for your comment. We have added the following sentence in the methodology for further clarification:

'The percentage share of PM10 in the aerosol mixture was calculated as $100*PM10/ (PM10 + PM_{pollen})$'.

We have also included the mean PDR value for each pollen type considering all cases. The following sentences have been added to the manuscript:

Line 243: 'Considering all cases with the variable birch contribution to the aerosol mixture a mean PDR of $4 \pm 2\%$, $16 \pm 6\%$, $13 \pm 8\%$ and, $18 \pm 8\%$ at 355, 532, 910 and 1565 nm wavelengths was estimated, respectively.'

Line 283: '….and a mean PDR of $4 \pm 2\%$, $25 \pm 15\%$, $14 \pm 9\%$ and, $21 \pm 6\%$ at 355, 532, 910 and 1565 nm wavelengths was obtained, respectively.'

**In line 268~271 and Figure 5, you can see the difference between shang et al (2022), but there is no difference between Bolnmann (2019, 2021) and this study's PDR532, so it would be nice to add it in Figure 5**.

Thank you for your suggestion. Figure 5 represents the characteristic PDR behaviour of birch pollen with minimum contribution of other aerosols at high pollen counts thus, a straightforward comparison is not necessarily valid. One way to include Bohlmann et al., (2019, 2021) works in Figure 5 is to estimate the spectral dependence of PDR at the birch concentration and PM10 share reported in these two studies. Although our dataset includes a wide range of birch pollen concentration and PM10 percentage shares, a close enough pair combination of birch pollen amount and PM10 percentage share to the ones reported at Bohlmann et al., (2019, 2021) works was not found. This means that, we can either report the spectral dependence of PDR for the pollen concentration or the PM10 share found in the aforementioned studies. Having this is mind, we have included here an adaptation for the pollen concentration, and we discuss the implications of this approach.

To make the comparison as straightforward as possible, we have considered three case studies as reported at Bohlmann et al., (2019, 2021) works. For these cases, we know the exact amount of birch to other pollen types and the full optical profiles are available. In turn, this ensures that the slightly different first layer definition between our studies can be modified to adapt to this manuscript's definition. The three chosen case studies are on the 6th of May 2016 (see Fig. 3 in Bohlmann et al., (2019)) (case 1), the 16th of May 2021(case 2) and the 17th of May 2021 (see Fig. 4 in Bohlmann et al., (2021)) (case 3). These cases were chosen because they had a 100% birch pollen contribution with no presence of pine/spruce. Regarding the birch pollen concentration, a 205 #/m$^3$ (case 1), 3247 #/m$^3$ (case 2) and 3226 #/m$^3$ (case 3) were estimated using the in-situ pollen collector. Although the PM10 share in the aerosol mixture is not reported in the original papers, we have calculated them here. The PM10 share was 86 % (case 1), 10 % (case 2) and 16 % (case 3), respectively with an actual PM10 concentration of 9 μg/m$^3$ (case 1), 2 μg/m$^3$ (case 2) and 4 μg/m$^3$ (case 3), respectively. The mean RH amounted to about 53 % (case 1), 36 % (case 2) and 39 % (case 3), respectively. Using Figure 4, we have estimated the PDR at the birch concentration occurred during the three cases. Note that cases 2 and 3 had a similar birch concentration hence one estimation was enough. We conclude that the concentration adaptation forecasts the PDR values with satisfactory results (good correlation in two out of three cases). Cases 2 and 3 had similar birch concentration and although the PDR in case 3 was higher the PM10 share was also higher. In the updated Figure 5 (see below), we see that already at ~3200 #/m$^3$ with a 16% PM10 share contribution, the characteristic PDR is concluded. Reading Figure 4 from the PM10 share perspective, the 9 and 16% PM10 share in cases 2 and 3 imply a mean PDR of 5 ± 2%, 20 ± 7% and, 25 ± 4% at 355, 532 and 1565 nm wavelengths, respectively. This range of values covers the spectral PDR behavior in both case 2 and 3. To shed further light, we have included the mean volume aerosol size distributions for these two cases. It is evident that case 3 had a notable difference in the aerosol population above 2.5 μm in size compared to case 2. No mineral dust was present in the atmosphere and the BC concentration was the same between the two cases. We conclude that the pollen concentration or PM10 share adaptation is a simplified approach. The difference between case 2 and case 3 may lie in the additional presence and type of pollen fragments with a diameter less than 10 μm. In turn, this marks the importance of instrument synergies and, at the same time, points out the complexity in pollen detection and classification using lidars. It also brings up the importance of pollen fragment detection.

[Figure]

Volume aerosol distributions for the 16th and 17th of May 2019.

[Figure]

Updated Figure 5 including Bohlmann et al., (2019 & 2021) works.

Given that Figure 5 represents the characteristic birch PDR spectral dependence, adding individual cases with variable contribution of birch pollen and PM10 aerosols can rather confuse the reader and shift the focus of this figure. Therefore, we prefer to leave Figure 5 as it is.

**line 28, SSPs -> SPPs**

Corrected

**line 61, bioaresols -> bioaerosols**

Corrected

**line83, Stremline -> Streamline**

Corrected

---

## Author Comment (AC3)

*The authors would like to thank the reviewer for his/her valuable comments and suggestions. We have modified the manuscript with the proposed changes along with step-by-step answers to the suggestions. Please note that changes have been highlighted (in bold or 'track changes') in the manuscript and the corresponding answers to the reviewer by text below. The original comments are presented in bold letters.*

Reviewer #1

**Today it is well accepted that the pollen play an important role in the process of aerosol cloud interaction. Still, the pollen optical properties, as well as the dynamic of their spatio – temporal variations are not studied sufficiently. One of the reasons is that pollen are normally mixed with other types of aerosol, and it is difficult to characterize the properties of "pure" pollen. From this point of view, the study presented is very important. The authors use three remote sensing instruments and provide the particle depolarization ratios at four wavelengths, from UV to IR. It is also important that authors consider numerous measurement cases, accumulated during field campaigns, thus allowing to estimate the depolarization ratio of pure birch and pine pollen. The paper is well and clearly written and is suitable for ACP. I have just several technical notes.**

**Notes:**

**Decrease of depolarization at 900 nm looks unexpected. But considering uncertainty of the measurements, may be this decrease is inside the uncertainty interval**.

Thank you for the comment. Indeed, there is a slight decrease observed at 910nm wavelength under the assumption that the theoretical molecular depolarization ratio is 0.0038. Unfortunately, limitations in the signal-to-noise ratio (SNR) did not allow us to retrieve the experimental molecular depolarization ratio in clean atmosphere thus the assumption of a 100% departure was used as an alternative. Under this scenario the particle depolarization ratio was 3% higher which was added on top of the standard deviation of the observations themselves. We do plan to fully characterize the noise in CL61 instruments and proceed with the uncertainty calculation of the depolarization retrievals thus reduce the PDR uncertainty, but this is the scope of a forthcoming paper.

**I wonder, why in Fig.2a pollen concentration is shown in logarithmic scale. Probably variations would be better seen in linear scale. But this is up to authors**.

We agree with the reviewer that the linear scale depicts the daily variation better, but we have chosen the logarithmic scale since the 'rest' category presented very low concentrations and therefor was only visible with the logarithmic scale. For clarification, below is the same figure with linear scale. Nevertheless, we prefer the logarithmic scale for the reason mentioned above.

[Figure]

**Fig.2b. I would change scale of backscattering to 0 - 2 Mm-1sr-1, to see details of profiles. The layer looks to be well mixed and backscattering at 532, 910, 1064 does not change with height, while at 355 nm it increases below 1 km. Can it be effect of overlap? This increase correlates with drop of particle depolarization at 355**.

We have updated Figure 2c to reviewer's suggestion. Regarding the profile of the backscatter coefficient at 355nm, that is a product of the combination of particle backscatters between the far field and the near field channels. This means that we perform the Klett solution independently at both channels and then merge the products. The merging region in this case realized at 1500m. Below this height the particle backscatter comes from the near field signal alone while above this height the far field observations were used. This is our standardized solution to correct for the overlap at the far field channel. The near field channel has an overlap of about 120m. At the moment, we do not perform an overlap correction in the near field channels as we consider observations starting a few hundred meters above the full overlap of the near field. This is not the case for the retrieval of the particle depolarization ratio (PDR). For the PDR we use the merged particle backscatter and since the depolarization channel is only present at the far field, the volume depolarization ratio (VDR) is affected by the different overlaps of the 355nm near-far field channels limiting the usefulness of the signal in the lowest part.

**Fig.2c. I would change scale of BAE also: -1 – 2**.

We have updated Figure 2d to reviewer's suggestion.

**Fig.3. If dust contribution increase, it should increase also depolarization at 355 nm. Was it observed**?

Due to the lower limit restriction at 355nm wavelength, the particle depolarization ratio is available at 800m above ground level. Below that height, we assume well mixed conditions and

the value at 800m is assumed to be indicative all the way down to the surface. This assumption may be valid during well mixed conditions under a convective boundary layer, but it is not necessarily valid at other times. Therefore, we didn't observe an increase at 355nm particle depolarization ratio (PDR) in the first layer. Given the low aerosol optical depth (AOD) projected by the models, mineral dust may have a minute contribution, if at all, to the AOD of the first layer. If mineral dust is present, it is certainly mixed with birch pollen and smoke as indicated by the in-situ observations.

**Line 333. Did authors estimate the EAE value**?

Given the location of the measurement site and the time of the year, the solar background radiation restricts Raman retrievals and therefore the independent estimation of the extinction coefficient is not widely available. We have only a few Raman cases during this period thus a statistical approach similar to what we have done is not feasible. Accounting for the lower boundary top heights during nighttime, the overlap limitation of the instrument and the aerosol conditions which might not be the same as during daytime (see Fig.3) a straightforward connection is therefore not necessarily valid.